# Agarose-Degrading Characteristics of a Deep-Sea Bacterium Vibrio Natriegens WPAGA4 and Its Cold-Adapted GH50 Agarase Aga3420

**DOI:** 10.3390/md20110692

**Published:** 2022-11-01

**Authors:** Mengyuan Zhang, Jianxin Wang, Runying Zeng, Dingquan Wang, Wenxin Wang, Xiufang Tong, Wu Qu

**Affiliations:** 1Marine Science and Technology College, Zhejiang Ocean University, Zhoushan 316000, China; 2Third Institute of Oceanography, Ministry of Natural Resources, Xiamen 361000, China

**Keywords:** *Vibrio natriegens*, deep sea, agarase, GH50, cold-adapted

## Abstract

Up until now, the characterizations of GH50 agarases from *Vibrio* species have rarely been reported compared to GH16 agarases. In this study, a deep-sea strain, WPAGA4, was isolated and identified as *Vibrio natriegens* due to the maximum similarity of its 16S rRNA gene sequence, the values of its average nucleotide identity, and through digital DNA–DNA hybridization. Two circular chromosomes in *V. natriegens* WPAGA4 were assembled. A total of 4561 coding genes, 37 rRNA, 131 tRNA, and 59 other non-coding RNA genes were predicted in the genome of *V. natriegens* WPAGA4. An agarase gene belonging to the GH50 family was annotated in the genome sequence and expressed in *E. coli* cells. The optimum temperature and pH of the recombinant Aga3420 (rAga3420) were 40 °C and 7.0, respectively. Neoagarobiose (NA2) was the only product during the degradation process of agarose by rAga3420. rAga3420 had a favorable stability following incubation at 10–30 °C for 50 min. The *Km*, *Vmax*, and *kcat* values of rAga3420 were 2.8 mg/mL, 78.1 U/mg, and 376.9 s^−1^, respectively. rAga3420 displayed cold-adapted properties as 59.7% and 41.2% of the relative activity remained at 10 3 °C and 0 °C, respectively. This property ensured *V. natriegens* WPAGA4 could degrade and metabolize the agarose in cold deep-sea environments and enables rAga3420 to be an appropriate industrial enzyme for NA2 production, with industrial potential in medical and cosmetic fields.

## 1. Introduction

Agar is the main component in the cell walls of red algae [1]. As the skeleton structure of agar [2], agarose possesses alternately repeated units of β-D-galactose and 3,6-anhydro-L-galactose [3]. Agarose can be degraded into oligosaccharides and monosaccharides with the potential for anti-inflammatory, anti-cancer, anti-caries, whitening, and moisturizing bio-activities in the industries of medicine, health products, cosmetics, and energy [4]. In addition, the polysaccharides (including cellulose, chitin, and agarose) are the carbon sources in environments, and the degraders of these polysaccharides are also participants in the carbon cycle [5,6]. Therefore, the degradation of agarose is of great value in the studies of industry production [5] and the carbon cycle [7,8].

Microbial degradation is the ideal method for the industrial degradation of agarose due to its high efficiency, environmental protection, energy conservation, and stable reaction product [9]. Agarases, including α- and β-agarase [1], are the molecular fundamentals of agarose-degrading microbes. α-agarase, belonging to the GH96 family, cuts the α-1,3-glycosidic bond and degrades agarose into agaro-oligosaccharides (AOS) with 3,6-anhydro-L-galactose at the reducing ends of the product. β-agarase, belonging to the GH16, GH39, GH50, GH42, GH86, and GH118 [10] families [1,11,12], cuts the β-1,4-glycosidic bond and degrades agarose into neoagaro-oligosaccharides (NAOS) with β-D-galactose at the reducing ends of the product [3,13]. To date, most agarases are β-type [14]. Of these β-agarases, GH42 [11] and GH50 [15] agarases are exo-types that only produce neoagarobiose (NA2). GH16 [16], GH39 [12], GH86 [17], and GH119 [10] agarases are endo-types that produce NAOS with mixed degrees of polymerization, including NA2, neoagarotetraose (NA4), neoagarohexaose (NA6), neoagarooctaose (NA8), neoagarodecaose (NA10), and neoagarododecaose (NA12).

*Vibrio* species are well-known utilizers of complex organic carbohydrates. They are also widely reported in agarose-degrading activities [18,19,20]. To date, GH16 agarases from *Vibrio* species (http://www.cazy.org/IMG/krona/GH16_krona.html, accessed on 21 July 2022) are well characterized [5,21,22], whereas the characteristics of other β-agarase families in the genus *Vibrio* are rarely studied, according to records in the CAZy database (http://www.cazy.org/IMG/krona/GH50_krona.html, accessed on 21 July 2022). In this study, a bacterium, *Vibrio natriegens* WPAGA4, with agarase activity was isolated from deep-sea sediments. Subsequently, a cold-adapted GH50 agarase was expressed and characterized in the genome sequence of *V. natriegens* WPAGA4 to explore its potential value in industrial applications and ecological functions.

## 2. Results

### 2.1. Identification and Taxonomy of Strain WPAGA4

Strain WPAGA4, showing a white, circular, and smooth colony, was isolated from the deep-sea sediments in this work. The 16S rRNA gene sequence of WPAGA4 has the highest similarities, of 99.12% and 99.26%, with that of *V. natriegens* NBRC 15636 (= ATCC 14048 = DSM 759) in the Ezbiocloud and NCBI databases, respectively. According to the phylogenetic analysis based on the genome sequences (Figure 1A,B), WPAGA4 showed the closest genetic distance to *V. natriegens* NBRC 15636 (Figure 1C). The genome of WPAGA4 had the highest DNA–DNA hybridization (DDH) and average nucleotide identity (ANI) values of 84.6 and 98.23, respectively, with *V. natriegens* NBRC 15636 (Appendix A). Therefore, this strain was assigned to *V. natriegens*, namely, *V. natriegens* WPAGA4.

### 2.2. The Determination and Optimization of the Agarose-Degrading Activity in the Broth

The optimum culture temperature for the agarose-degrading activity in the culture broth of *V. natriegens* WPAGA4 was 31 °C (Figure 2A). Additional agar improved the agarose-degrading activity to a certain extent (Figure 2C). Furthermore, additional amylum and lactose improved the agarose-degrading activity, whereas glucose, sucrose, maltose, and glycerin could inhibit the activity in WPAGA4 broth (Figure 2C). The additional nitrogen sources presented in this work could not increase the agarose-degrading activity in the broth. Notably, NH4NO3 and urea obviously inhibited the agarose-degrading activity in the broth of WPAGA4 (Figure 2D). The maximum agarose-degrading activity of *V. natriegens* WPAGA4 in the fermentation broth was 19.49 U/mL.

### 2.3. Analyses of the Genome Sequence and Putative Agarase Gene Aga3420 of Strain WPAGA4

Two circular chromosomes in *V. natriegens* WPAGA4 containing 3,241,580 bp and 1,994,952 bp were assembled (Figure 1A,B). A total of 4561 coding genes, 37 rRNA, 131 tRNA, and 59 other non-coding RNA genes were predicted in the genome of *V. natriegens* WPAGA4, and 5 CRISPRs, 12 gene islands, and 1 prophage were annotated in the genome sequence.

A total of 173 carbohydrate-active enzymes genes were annotated in the genome of *V. natriegens* WPAGA4, including 16 auxiliary activity genes, 26 carbohydrate-binding modules, 15 carbohydrate ester genes, 75 glycoside hydrolase genes, 34 glycosyltransferase genes, 7 polysaccharide lyase genes (Figure 1C). Four putative GH50 agarase genes, including aga3418, aga3419, aga3420, and aga3421, were annotated in the genome sequence based on the dbCAN result (Appendix A and Appendix A).

### 2.4. Expression and Characterization of the Recombinant Agarases

Among the four recombinant agarases, only the recombinant Aga3420 (rAga3420) showed obvious agarase activity under the conditions described in Section 4.6. Therefore, rAga3420 was selected for further analysis. The phylogenetic analysis showed that the amino acid sequence of Aga3420 had the closest genetic distance to *Agarivorans gilvus* agarase (accession number: AQT38174.1) (Figure 3).

rAga3420 with a molecular weight of ~100 kDa, which was consistent with its theoretical molecular weight (104.87 kDa), was purified based on its 6xHis tag in this work (Figure 4A). The optimum temperature of rAga3420 was 40 °C (Figure 4B). In addition, 59.7% and 41.2% of the relative rAga3420 activity remained at 10 °C and 0 °C, respectively (Figure 4B). The activity of rAga3420 is rapidly lost at temperatures higher than 40 °C (Figure 4B). The optimum pH of rAga3420 was 7.0 (Figure 4C). rAga3420 had good stability after the incubation at 10–30 °C for 50 min (Figure 4D); however, 22.3% of the relative activity was lost after the incubation at 30 °C for 60 min (Figure 4D). The characterization comparison of rAga3420 and other GH50 agarases are listed in Appendix A. The metal ions in this work could improve the rAga3420 activity (Figure 4E). Dithiothreitol (DTT) could increase the activity of rAga3420, whereas sodium dodecyl sulfate (SDS) and Zn^2+^ inhibited its activity (Figure 4F). Salinity had an obvious negative effect on rAga3420 activity (Figure 4G). The values of *Km*, *Vmax*, and *kcat* were 1.5 ± 0.6 mg/mL, 53.8 ± 1.4 U/mg, and 259.2 ± 21.5 s^−1^, respectively, according to the Lineweaver—Burk plot (Appendix A).

### 2.5. Agarose Degradation Products by rAga3420

The degradation products of agarose by rAga3420 were detected according to the mass-to-charge ratio (m/z) values. The agarose was only decomposed into the oligosaccharides with the degree of polymerization of 2 ([NA2 + H]^+^, 325 m/z; [NA2 + Na]^+^, 347 m/z; Figure 5) at the degradation times of 5 min (Figure 5A), 10 min (Figure 5B), 20 min (Figure 5C), 30 min (Figure 5D), 60 min (Figure 5E), 120 min (Figure 5F), and 240 min (Figure 5G). NA2 is the only oligosaccharide product during the degradation process, indicating rAga3420 is an exo-type agarase.

## 3. Discussion

Polysaccharide degradation is an important component of the carbon cycle in marine environments [23], including the deep sea [24]. Microorganisms are crucial players in driving polysaccharide degradation, which participates in the carbon cycle of deep-sea environments. The genus *Vibiro*, such as *V. diabolicus* [25], *V. profundi* [26], *V. bathopelagicus* [27], and other species that remain unidentified [28,29], have been widely isolated in the deep sea. Of them, several strains were reported with polysaccharide-degrading activities. For example, *Vibrio* sp. JAM-A9m, isolated from the deep sea, was detected with alginate lyase activity [29], and another deep-sea *Vibrio* strain showed cold-adapted amylase activity [30] in previous works. Therefore, the genus *Vibrio,* from the deep-sea environment, is a contributor to polysaccharide degradation. This work reported the first *Vibrio* strain isolated from the deep-sea environment with agarose-degrading activity. Although the agarase activity of *V. natriegens* WPAGA4 (19.49 U/mL) was moderate among the agarose-degrading strains, such as *Agarivorans* sp. JAMB-A11 (10.3 U/mL) [31], *Vibrio* sp. PO-303 (63.6 U/mL) [32], and *Pseudomonas* sp. SK38 (32.3 U/mL) [33], the current study indicated that this strain was potentially involved in the polysaccharide degradation and carbon cycle processes inside the deep-sea environment due to its agar-degrading ability.

As far as we know, the agarose concentration in deep-sea environments has not been reported, and the reason that an agarose-degrading strain exists in the deep sea remains unexplored until now. However, the WPAGA4 strain is not the only strain isolated from the deep-sea environment with agar-degrading activity. In former studies, deep-sea strains, including *Flammeovirga* sp. OC4 [13,16], *Shewanella* sp. WPAGA9 [16], *Flammeovirga pacifica* WPAGA1 [34], *Microbulbifer* sp. JAMB A3, *Microbulbifer* sp. JAMB A7, *Microbulbifer* sp. JAMB A24, *Microbulbifer* sp. JAMB A33, *Microbulbifer* sp. JAMB A94, *Microbulbifer* sp. JAMM 0654, *Microbulbifer* sp. JAMM 0793, *Microbulbifer* sp. JAMM 1327, and *Microbulbifer* sp. JAMM 1340 [35], were isolated and detected with agar-degrading activity. In the least, the presence of these strains indicates that agar degradation might potentially be a part of the carbon cycle process within the deep-sea environment.

Although *V. natriegens* isolated from other environments have been used for agarase production in several studies, the agarose-degrading activity and the agarase of *V. natriegens* have been studied less [36,37]. The current work characterized a GH50 agarase, Aga3420, which supported the agarose-degrading activity of *V. natriegens* WPAGA4. Similar to other GH50 agarases, rAga3420 was an exo-type that degraded agarose into NA2 [1,38], which is an essential step for microorganisms to metabolize agarose as energy material [39]. As far as we know, rAga3420 is the first GH50 agarase from the deep sea with characterized enzyme properties, including optimum temperature, pH, and thermostability. Compared with the GH50 agarases from other environments, rAga3420 possessed outstanding cold-adapted properties (Appendix A). For example, two GH50 agarases, AgaA50 and AgaC50, in *Microbulbifer elongatus* PORT2 isolated from Indonesian coastal seawater lost ~80% of its relative activity at 30 °C [40], and GH50 agarase Aga575 from the surface of fresh porphyra only retained <40% of its relative activity at 25 °C [15]. However, rAga3420 could maintain more than 40% and 50% of the relative activities at 0 °C and 10 °C, respectively, thereby ensuring that *V. natriegens* WPAGA4 could use agarose and agar as energy materials at low temperatures and could thus drive the polysaccharide cycle in the harsh, cold environment of the deep sea [41].

Recently, scientific interest in cold-adapted enzymes has been increasing [42]. Cold-adapted enzymes can work at low and moderate temperatures (<40 °C), thereby saving energy expenditure and equipment optimization costs in the heating process [43]. In addition, the by-products of the reaction can be reduced at low temperatures [44]. The cold-adapted rAga3420 in this work could serve as an energy- and cost-saving bio-source for industrial agarose degradation and oligosaccharide production [12]. Meanwhile, low temperatures could also prevent caramelization, a potential by-product of high-temperature polysaccharide degradation [45]. Therefore, the agarase rAga3420, with its low temperature activity, is not only a potential driver of the polysaccharide cycle, but it is also an ideal bio-tool for oligosaccharide production in industrial applications [12].

Although the NAOS, with different degrees of polymerizations, such as NA2, NA4, NA6, and NA8, are reported as active substances [46], NA2 has its advantages in several bio-activities. For example, the antioxidant activity of NA2 is better than that of NA4 and NA6 as it can rapidly remove 2,2-diphenyl-1-picrylhydrazyl [47]. Therefore, the preparation and production of NA2 is important for the development of NAOS industries. However, the endo-type agarases that produce NA2 degrade agarose into mixture products [12], thereby increasing the cost of purification. The exo-type GH50 agarase rAga3420 only produces NA2 during the degradation process. The single product of rAga3420 lowered the cost and reduced process complexity for the purification and production of NA2. In addition, the high optimum temperatures of agarases [13,48,49] increase the energy input during NAOS production. The cold-adapted properties of rAga3420 showed an optimum temperature of 40 °C and retained more than 70% of the relative activity at 20 °C and 30 °C, indicating that rAga3420 could produce NA2 at room temperature. These properties reduced the energy costs of NA2 production. These characterizations make rAga3420 an ideal bio-tool for agarose degradation and NA2 production.

A total of four putative agarase genes were expressed in this work; however, only rAga3420 was detected to have the agarase activity under the conditions provided in this work. We attributed this to the false-positive prediction, minimal expression, protein misfold, and incongruous reaction conditions and substrates in the former work [50]. Nevertheless, it still reminds us that there are several unknown agarase resources in this strain and in deep-sea environments, which remain to be explored in future works.

In conclusion, a *V. natriegens* WPAGA4 strain with agarose-degrading activity was isolated from deep-sea sediments in the current work. A GH50 agarase gene, aga3420, was sequenced in the genome of *V. natriegens* WPAGA4 and was expressed in *E. coli* cells. The exo-type rAga3420 showed a cold-adapted property that maintained ~40% of the relative activity at 0 °C. This property ensured that *V. natriegens* WPAGA4 could degrade and metabolize agarose in a cold, deep-sea environment, and enable rAga3420 to be an appropriate industrial enzyme for NA2 production with further industrial potential in medical and cosmetic fields.

## 4. Materials and Methods

### 4.1. Isolation, Purification, and Identification

The deep-sea sediment sample was collected from the Western Pacific at site CM3MC04-1 (157°24′31″ E, 19°30′30″ N; 5378 m deep) in July 2007 using a core sampler. According to the practical guide [51], ~2 cm sediments were removed from the outer layer of the sample with a sterilized knife. Then, only the core part was stored in sterile centrifuge tubes which were stored in a −80 °C ultra-low temperature freezer. After the samples were transported back to the labs, they were serially diluted using the liquid 2216E medium (5.0 g of peptone, 1.0 g of yeast extracts, 0.01 g of ferric phosphate, and 1 L sea-water; pH 7.2–7.6) on a sterilized bench. The diluent was spread onto the 2216E plates, and the plates were cultured at 28 °C. The single colony, with agar collapse, was picked and streaked twice on the 2216E plates for further purification. 

A colony of the WPAGA4 strain was purified using the above method. The genomic DNA of WPAGA4 was extracted using an EasyPure Bacteria Genomic DNA Kit (Transgen, Beijing, China) according to the instructions provided. Its purity and integrity were tested with Nanopore 2000 (Thermo, Waltham, MA, USA) and 1% agarose gel electrophoresis, respectively. The high-quality genomic DNA was used as the template for the 16S rRNA gene amplification with the primer pairs of 27F (5’-AGAGTTTGATCCTGGCTCAG-3’) and 1492R (5’-TACGACTTAACCCCAATCGC-3’) according to the PCR conditions stated in previous work [52]. The PCR product was sequenced by Sangon Biotech (Shanghai, China) Co., Ltd. The sequence of the 16S rRNA gene was aligned to the databases of NCBI (https://www.ncbi.nlm.nih.gov/, accessed on 21 July 2022) and Ezbiocloud (https://www.ezbiocloud.net/, accessed on 21 July 2022). 

### 4.2. Genome Sequencing and Phylogenetic Analysis of Strain WPAGA4

The genome of strain WPAGA4 was sequenced by Beijing Biomarker Technologies (Beijing, China) using the platforms of Nanopore GridION (Oxford Nanopore Technologies, Oxford, UK) and Illumina Hiseq (Illumina, California, CA, USA). Canu v1.5 (http://canu.readthedocs.io/en/latest/ind-ex.html, accessed on 5 January 2022) [53] was used for the assembly of high-quality reads. Subsequently, the genome sequence was further corrected by using Pilon v1.22 software (https://github.com/broad-institute/pilon/releases/download/v1.22/pilon1.22.jar, accessed on 5 January 2022) [54]. Circlator v1.5.5 (https://github.com/sanger-pathogens/circlator, accessed on 5 January 2022) [55] was used for the cyclization of the genome. Finally, a complete and circular genome of WPAGA4, without any gaps, was generated. The sequence of the WPAGA4 genome was submitted to the Genome-to-Genome Distance Calculator (https://ggdc.dsmz.de/home.php, accessed on 21 July 2022) [56] to perform the genome-based phylogenetic analysis and calculate the DDH value. The value of ANI was calculated by the website (https://www.ezbiocloud.net/tools/ani (accessed on 21 July 2022) [57]. The coding genes in the genome were predicted by using Prodigal 2.6.1 (https://github.com/hyattpd/Prodigal, accessed on 5 January 2022) [58]. The genes of non-coding RNA, rRNA, and tRNA were predicted by using tRNAscan-SE 1.3.1 (http://lowelab.ucsc.edu/tRNAsca-n-SE/, accessed on 5 January 2022) [59], Infernal 1.1 (https://launchpad.net/ubuntu/+source/infern-al/1.1.1-1, accessed on 5 January 2022) [60], and the Rfam database (http://rfam.xfam.org/, accessed on 21 July 2022) [61], respectively. CRT 1.1 (http://www.room220.com/crt/, accessed on 5 Jan 2022) [62], IslandPath-DIMOB v4.0 (https://www.pathogenomics.sfu.ca/islandviewer/resources/, accessed on 5 January 2022) [63], and PhiSpy 4.2.19 (https://anaconda.org/bioconda/phis-py, accessed on 5 January 2022) [64] were used for the prediction of CRISPR, gene islands, and prophages, respectively. The functions of the coding genes were annotated by using BLAST software (https://blast.ncbi.nlm.nih.gov/Blast.cgi, accessed on 5 January 2022) against the NCBI nr database (ftp://ftp.ncbi.nih.gov/blast/db, accessed on 5 January 2022). The CAZyme genes were annotated using the dbCAN2 meta server (https://bcb.unl.edu/dbCAN2/index.php, accessed on 5 January 2022) [65]. The genome sequence of WPAGA4 was deposited into the GenBank database under the accession numbers CP094880 and CP094881.

### 4.3. The Determination and Optimization of the Agarase Activity in the Fermentation Broth 

To optimize the agarose-degrading activity in the fermentation broth of WPAGA4, the strain cells were cultured in 50 mL liquid 2216E medium at 25 °C, 28 °C, 31 °C, 34 °C, and 37 °C for 24 h with an inoculation size of 1% and a rotation speed of 200 rpm for the 250 mL shake flask. Then, the culture broth was centrifugated at 6000 rpm for 30 min at 4 °C, and the supernatant was collected for the determination of agarase activity. The methods for the measurement and analysis were reported in our previous work [40]. To measure the substrate’s inducibility, the cells were cultured at 31 °C for 24 h in the blank 2216E medium and the 2216E medium with an additional 0.2% agarose (Westbio, Spanish). Then, the supernatant was collected, and the enzymic activity was determined as mentioned above. To determine the effects of the carbon and nitrogen sources on the supernatant activity of WPAGA4, 1% (*w/v*) of NH_4_NO_3_, KNO_3_, NH_4_Cl, (NH_4_)_2_SO_4_, urea, corn steep liquor, soybean cake, glucose, sucrose, maltose, glycerin, amylum, and lactose were added to different samples of 2216E media used in the strain culture. The culture was performed at 31 °C for 24 h, and the supernatant was collected for agarase activity determination as mentioned above. All determinations were repeated three times.

### 4.4. The Analysis of Agarase Gene Aga3420 in the Genome of WPAGA4

An agarase gene, aga3420, was annotated in the genome of WPAGA4. The nucleoti-de sequence of aga3420 was aligned using BLAST software against the NCBI nr data-base for the similarity analysis. The analysis of the glycoside hydrolase family of the putative agarases was performed by the dbCAN2 meta server (https://bcb.unl.edu/dbCAN2/index.php, accessed on 5 January 2022) [65]. Phylogenetic analysis of the amino acid sequence of aga3420 was conducted using Mega 5.0 software with the neighborhood-joining method.

### 4.5. The Cloning and Expression of the Agarase Genes

Primer pairs in Appendix A were used for the amplification of the putative agarase genes aga3420 using a C1000 Thermal Cycler (Bio-Rad, California, CA, USA). The PCR program was as follows: 3 min at 95 °C; 25 cycles of 95 °C for 15 s; 55 °C for 15 s; 72 °C for 1 min/kb; and an extension at 72 °C for 6 min. The PCR products were further purified using a Universal DNA Purification Kit (Tiangen Biotech, Beijing, China). The purified PCR products were ligated with the pEAZY^®^-Blunt E2 expression vector (TransGen, Beijing, China). The recombinant vector was transformed into the competent cells of *E. coli* BL21(DE3) pLysS.

The *E. coli* BL21(DE3) pLysS cells harboring the recombinant vector were cultured at 37 °C in LB broth (5.0 g yeast extracts, 10.0 g peptone, 10.0 g NaCl, and 1 L deionized water) with additional 100 μg/mL ampicillin. The culture broth was induced by 0.1 mM IPTG at 16 °C for 18 h until the OD600 value reached 0.6–0.8. The cells were collected by centrifugation at 10,000 rpm for 10 min at 4 °C, and the cells were fractured using a sonicator (Scientz, Ningbo, China). The Ni-nitrilotriacetic acid (NTA) Sefinose™ Resin (Tiangen Biotech, China) was used to further purify rAga3420 with 6 × His-tagged from the crude enzyme solution. The molecular weight and purity of rAgaW1540 were subsequently estimated using SDS-PAGE and Coomassie brilliant blue staining.

### 4.6. Activity Determination and Characterization of rAga3420

The protein concentration was determined using the Bradford method according to the former study [66]. The agarase activity (U) was defined as the amount of the enzyme that released 1 μmol of reducing sugars per minute under certain conditions. The measurement method for the agarase activity was conducted according to the previous study [16]. Briefly, 10 μL of the enzyme solution (20.7 μg/mL) was mixed with 240 μL agarose (0.4%; *w/v*) dissolved in different pH buffers, including sodium citrate-dibasic sodium phosphate (50 mM; pH 3.0–8.0), Tris-HCl (50 mM; pH 8.0–9.0), and glycine-NaOH (50 mM; pH 9.0–10.6), and the reaction mixture was incubated at temperatures of 0 °C, 10 °C, 20 °C, 30 °C, 40 °C, 50 °C, 60 °C, 70 °C, and 80 °C for 20 min. Then, 750 μL 3,5-dinitrosalicylic acid (DNS) solution was added to the reaction mixture, and the reducing sugar was measured [67]. To determine the thermostability of rAga3420, the enzyme was pre-incubated at 10 °C, 20 °C, and 30 °C for 60 min, and the relative activity of rAga3420 was determined every 10 min. The effects of metal ions and chemical agents on rAga3420 were measured by determining the enzymatic activity in the solution containing 1 mM of Mg^2+^, Fe^2+^, Fe^3+^, Ca^2+^, Mn^2+^, Cu^2+^, Na^+^, K^+^, Zn^2+^, EDTA, SDS, DTT, and β-mercaptoethanol. The impact of salinity was determined by measuring the relative activity with 0.5–3.5 mmol/L of NaCl. The kinetic parameters of rAga3420 were measured by determining the activity within the agarose at final concentrations of 0.1–0.6 mg/mL, and the *Km* and *Vmax* values were calculated by the Lineweaver–Burk plot. All determinations were repeated three times. The relative activity was defined as the percentage of activity determined with respect to the maximum agarase activity.

### 4.7. The Determination of Degradation Products

The degradation products of agarose by rAga3420 were identified by liquid chromatography mass spectrometry (LC-MS; Agilent XEVO G2-XS QTOF, Palo Alto, PA, USA). According to the former work [68], the degradation solution was filtered with a filter membrane of 0.22 μm pore size and then mixed with a triple volume of absolute ethanol for removing the protein and undegraded polysaccharides. Subsequently, a C18 column (2.1 × 100 mm, 1.8 µm; Waters, Milford, MA, USA) was used for the separation of the oligosaccharides by gradient elution with 0.1% formic acid–water and formic acid–acetonitrile at a flow rate of 0.5 mL/min. The injection volume was 1 μL. The values of m/z from 50 to 3000 were analyzed by a quadrupole mass analyzer with positive-ion ESI mode. The interface voltage, detector voltage, curved desolvation line temperature, and heat-block temperature were set as 4.5 kV, 1.65 kV, 200 °C, and 200 °C, respectively.

## Figures and Tables

**Figure 1 marinedrugs-20-00692-f001:**
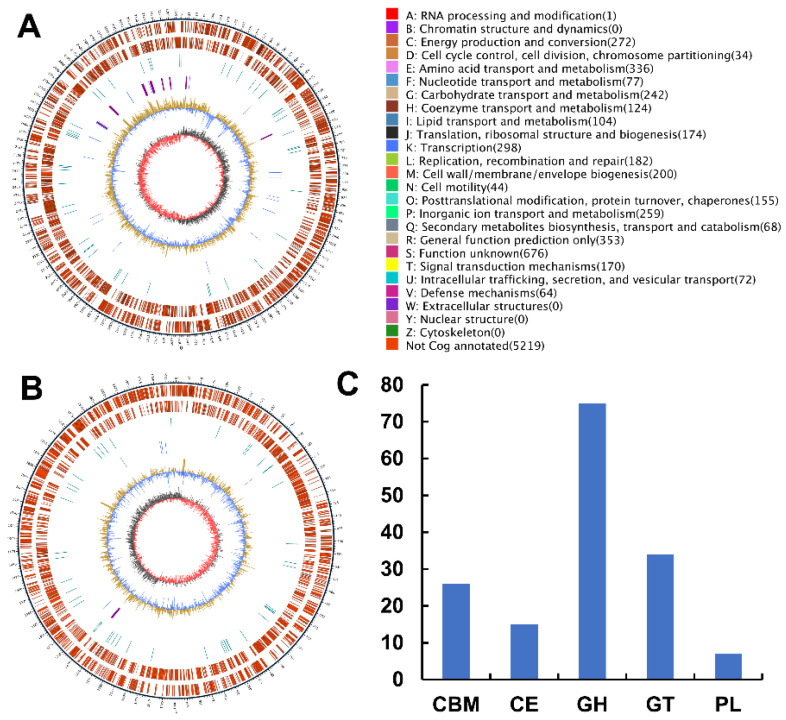
The genome analysis of *Vibrio natriegens* WPAGA4. The circular maps of the two chromosomes of *V. natriegens* WPAGA4 are shown in (**A**,**B**). The outer layer of the map is the size of the genome; the second and third circles are the genes on the positive and negative strands, respectively. The colors stand for the COG function classifications; the repetition sequences are shown in the fourth circle; the fifth circle is tRNA (blue) and rRNA (purple); the sixth circle represents the GC content. The light-yellow part indicates that the GC content of the region is higher than the average GC content of the genome, and the blue part indicates that the GC content of the region is lower than the average GC content of the genome; the innermost circle stands for the GC-skew. Dark gray represents the region with G content greater than C, and red represents the region with C content greater than G. The CAZyme gene annotation results are shown in (**C**).

**Figure 2 marinedrugs-20-00692-f002:**
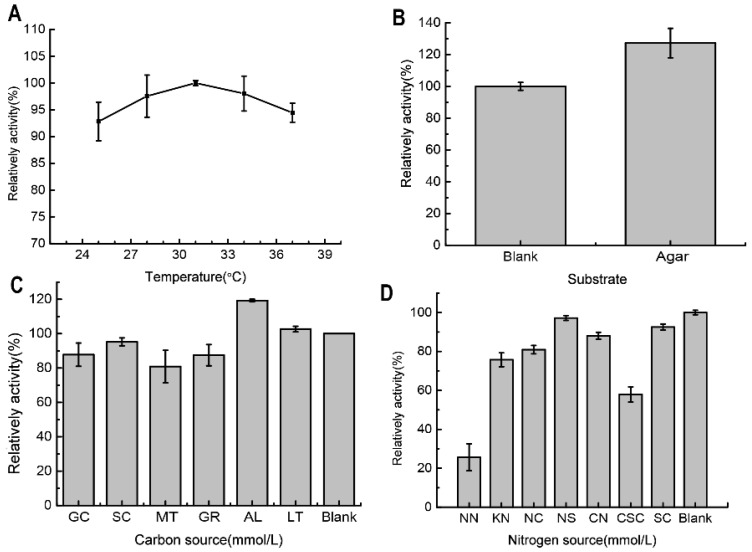
The optimization of the agarose-degrading activity in the fermentation broth of *V. natriegens* WPAGA4 is based on the fermentation temperature (**A**), agarose substrate (**B**), and carbon (**C**) and nitrogen sources (**D**). GC, Glucose; SC, Sucrose; MT, Maltose; GR, Glycerin; AL, Amylum; LT, Lactose; NN, NH4NO3; KN, KNO3; NC, NH4Cl; NS, (NH4)2SO4; CN, CH4N2O (urea); CSC, Com steep liquor; and SC, Soybean cake.

**Figure 3 marinedrugs-20-00692-f003:**
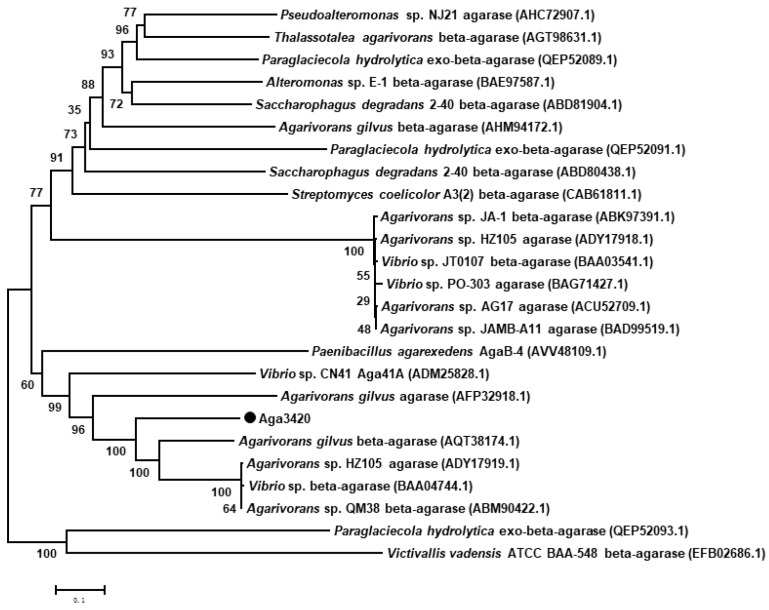
The phylogenetic analysis of the amino acid sequence of Aga3420.

**Figure 4 marinedrugs-20-00692-f004:**
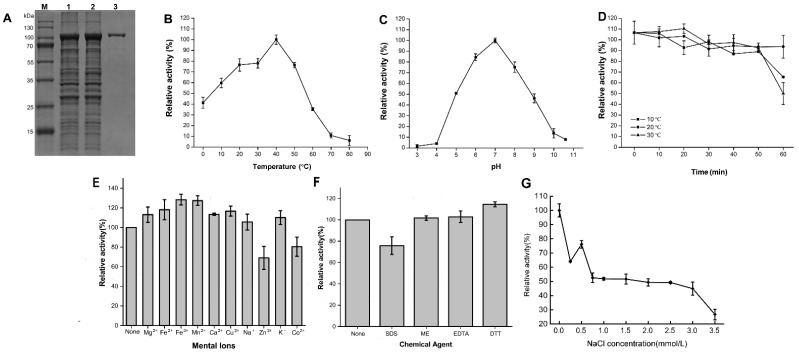
The expression and characterization of rAga3420. SDS-PAGE results for rAga3420 are shown in (**A**). Lanes M, 1, 2, and 3 represent the standard protein marker, non-induced cell lysis solution, IPTG-induced cell lysis solution, and purified rAga3420, respectively. The effects of temperature (**B**) and pH (**C**) on the relative activity of rAga3420. The stability (**D**) of rAga3420. The effects of metal ions (**E**), chemical agents (**F**), and salinity (**G**) on the relative activity of rAga3420. DTT, dithiothreitol; ME, β-mercaptoethanol.

**Figure 5 marinedrugs-20-00692-f005:**
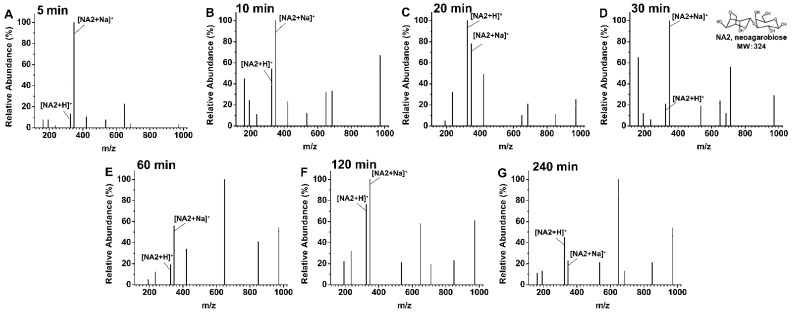
The degradation product of agarose by rAga3420 measured by LC-MS at the degradation times of 5 min (**A**), 10 min (**B**), 20 min (**C**), 30 min (**D**), 60 min (**E**), 120 min (**F**), and 240 min (**G**).

## Data Availability

The genome sequence of WPAGA4 was deposited into the GenBank database under the accession numbers CP094880 and CP094881.

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
