# Peer review of "Agarose-Degrading Characteristics of a Deep-Sea Bacterium Vibrio Natriegens WPAGA4 and Its Cold-Adapted GH50 Agarase Aga3420"

_marinedrugs, 2022, doi:10.3390/md20110692_

Round 1

Reviewer 1 Report

This study reports the isolation, genome sequencing and biochemical characterization of GH50 beta-agarase from a deep-sea marine bacterium.

Overall, this study lacks depth of analysis and general significance and novelty of findings. The shallow analysis of the genome is a shame since a deeper analysis and proper genome comparison of close relatives could bring up several interesting characteristics. The choice to express and shallowly characterize only one enzyme, when there are at least three other putative GH50 in the genome is also detrimental to the study and should be revised. The existing analysis have some flaws (see comments below) and some experiments needs to be repeated.

Specific comments and questions.  

1.

13: This first sentence should be removed since it falsely advertises novelty without any concrete qualifications.    

2.

37-40: This statement is hugely exaggerated and the reference that is quoted does not support it. This should be removed or rephrased.

3.

48-50: Same problem as above. This statement is not corroborated by the literature or data and should be removed.

4.

Figure 2. The caption should include explanations of all the abbreviations. Figure 2A: the size of the errorbars indicate no significant differences between temperatures. This experiment should be repeated with a higher amount of replicates or with a higher degree of thoroughness to decrease variance. Same goes for Figure2 B and C.  

5.

118: The dbCAN2 server should be quoted correctly in the method section.

6.

Figure 3: The phylogenetic analysis method has not been described and is normally performed on amino acid sequences. Should also at least include all characterized sequences listed in the CAZy-database

7.

131-132 and Figure 4: The results from EDTA vs. metal ions is contradicting themselves since EDTA chelates any bound metals in the enzyme. Metal experiment should be repeated with EDTA treated and then dialyzed enzyme to eliminate any previously bound metals.

8.

134-135: The kinetic parameters needs standard deviations and the Michalis-Menten curves needs to be shown (at least in the supplementary files).

9.

Figure 5: Why is the relative abundances changing irregularly through time and how can there be 100% of NA2+Na (panel A, B, C, E) and still other peaks visible at 20-60%? Relative abundance indicate a fraction of the total spectrum so the calculation is off. Furthermore Given the high kcat maybe a shorter timespan of 2-5 minutes should be included.

10.

163: Again, 25 characterized members of this family is hardly a rare observation and as described in line 174-180 it has been observed several times in relation to deep sea sediment. Please modify the language accordingly.

11.

184: So it has been studied already? Please quote the sources.

12.

214: Bad sentence, please rephrase.

13.

220: Why is cold adaptivity a positive trait in relation to industrial applications? Often industrial processes of viscous polymers favors high temperatures, where this enzyme would be ill suited. Please clarify.

14.

244-267: All tools needs to be cited correctly with the appropriate papers and not only the URL.

Author Response

Response:

Thank you for the valuable comments.

In our revised manuscript, the CAZymes in WPAGA4 were further analyzed, and a total of 173 carbohydrate-active enzymes genes were annotated in the genome of V. natriegens WPAGA4, including 16 auxiliary activity genes, 26 carbohydrate-binding modules, 15 carbohydrate ester genes, 75 glycoside hydrolase genes, 34 glyco-syltransferase genes, 7 polysaccharide lyase genes (Fig. 1C). Moreover, there are indeed other putative agarases in the genome; however, the recombinant proteins of these putative genes are found without any agarase activity. We contribute this result to a false annotation result; therefore, this data is not shown in the manuscript. 

The GH50 agarases in Vibrio species have not been well studied until now. The current work aims to provide a novel GH50 agarase from V. natriegens with cold-adapted and neoagarobiose-producing activities, which could serve as an ideal tool for oligosaccharide production industry. We have stated this novelty in lines 24-28, and 56-63.

Besides, the manuscript has been revised according to your specific comments. Please see the detail responses below. Many thanks again for your review and all the constructive comments which greatly improves our work.

Specific comments and questions. 

13: This first sentence should be removed since it falsely advertises novelty without any concrete qualifications.

Response: We have revised this sentence into “The characterizations of GH50 agarases from Vibrio species have rarely been reported until now”. This statement is based on the records in the CAZy database (http://www.cazy.org/IMG/krona/GH50_krona.html). Please also see our introduction section (lines 58-60).

37-40: This statement is hugely exaggerated and the reference that is quoted does not support it. This should be removed or rephrased.

Response: This sentence has been re-phrased into “Besides, the polysaccharides, including cellulose, chintin, and agarose, are the carbon sources in environments, and the degraders of these polysaccharides are also the participants in the carbon cycle [1]. Therefore, the degradation of agarose is of great value for the studies on industry production [1] and the carbon cycle [2,3]”, and the references that supported this statement has been supplemented. Please see lines 37-40.

References

[1] Hunt, D.E., Gevers, D., Vahora, N.M., and Polz, M.F. (2008). Conservation of the chitin utilization pathway in the Vibrionaceae. Applied and environmental microbiology 74, 44-51.

[2] Liu, Y., Jin, X., Wu, C., Zhu, X., Liu, M., Call, D.R., and Zhao, Z. (2020). Genome-wide identification and functional characterization of β-agarases in Vibrio astriarenae strain HN897. Frontiers in microbiology 11, 1404.

[3] Zhang, X., Lin, H., Wang, X., and Austin, B. (2018). Significance of Vibrio species in the marine organic carbon cycle—a review. Science China Earth Sciences 61, 1357-1368.

48-50: Same problem as above. This statement is not corroborated by the literature or data and should be removed.

Response: The statement has been removed from the revised manuscript.

Figure 2. The caption should include explanations of all the abbreviations. Figure 2A: the size of the errorbars indicate no significant differences between temperatures. This experiment should be repeated with a higher amount of replicates or with a higher degree of thoroughness to decrease variance. Same goes for Figure2 B and C.

Response: The abbreviation explanations have been supplemented in lines 105-107. We have repeated the experiments of Figure 2, and the variance has been decreased in the revised result.

118: The dbCAN2 server should be quoted correctly in the method section.

Response: The reference has been supplemented in line 310.

Figure 3: The phylogenetic analysis method has not been described and is normally performed on amino acid sequences. Should also at least include all characterized sequences listed in the CAZy-database

Response: Thank you for the valuable comment. We have revised the phylogenetic analysis based on the amino acid sequence, and the method has been described. Please see lines 117-120 and 310-311 in the revised manuscript.

  Now there are numbers of agarases in CAZy database. Therefore, a phylogenetic tree with so many sequences largely consumes the computing resources, and the huge tree would cover the target sequences. To improve this problem, many references [1-5] select several sequences from each agarase family to build the phylogenetic tree for a clearer result. We are sorry for unsatisfying your requirement, and we humbly ask if this result can be shown in the form as the former works.

References

[1] Dong C, Lin B, Song Y, et al. Characterization and activity enhancement of a novel exo-type agarase Aga575 from Aquimarina agarilytica ZC1[J]. Applied Microbiology and Biotechnology, 2021, 105(21): 8287-8296.

[2] Di W, Qu W, Zeng R. Cloning, expression, and characterization of thermal‐stable and pH‐stable agarase from mangrove sediments[J]. Journal of basic microbiology, 2018, 58(4): 302-309.

[3] Jung S, Lee C R, Chi W J, et al. Biochemical characterization of a novel cold-adapted GH39 β-agarase, AgaJ9, from an agar-degrading marine bacterium Gayadomonas joobiniege G7[J]. Applied microbiology and biotechnology, 2017, 101(5): 1965-1974.

[4] Cao S, Shen J, Zhang Y, et al. Expression and Characterization of a Methylated Galactose-Accommodating GH86 β-Agarase from a Marine Bacterium[J]. Journal of Agricultural and Food Chemistry, 2020, 68(29): 7678-7683.

[5] Xu Z X, Yu P, Liang Q Y, et al. Inducible expression of agar-degrading genes in a marine bacterium Catenovulum maritimus Q1T and characterization of a β-agarase[J]. Applied Microbiology and Biotechnology, 2020, 104(24): 10541-10553.

131-132 and Figure 4: The results from EDTA vs. metal ions is contradicting themselves since EDTA chelates any bound metals in the enzyme. Metal experiment should be repeated with EDTA treated and then dialyzed enzyme to eliminate any previously bound metals.

Response: Thank you for your strict review. The experiments have been repeated according to your comments. The effect of metal ions on the enzyme remains the same result with the former. However, EDTA shows no obviously effect on the agarase activity. Please see the revised manuscript (lines 134 and 136) and Figure 4.

134-135: The kinetic parameters needs standard deviations and the Michalis-Menten curves needs to be shown (at least in the supplementary files).

Response: The results has been revised in line 137-138 and the supplementary files (Fig. S1).

Figure 5: Why is the relative abundances changing irregularly through time and how can there be 100% of NA2+Na (panel A, B, C, E) and still other peaks visible at 20-60%? Relative abundance indicate a fraction of the total spectrum so the calculation is off. Furthermore Given the high kcat maybe a shorter timespan of 2-5 minutes should be included.

Response: The relative abundance of the highest peak in LC-MS was defined as 100% as the former studies’ definition [1-5]. The products under a shorter time (5 min) were determined in the revised manuscript. Please see Figure 5.

163: Again, 25 characterized members of this family is hardly a rare observation and as described in line 174-180 it has been observed several times in relation to deep sea sediment. Please modify the language accordingly.

Response: The sentence has been deleted (lines 168 and 169). Thank you for the comment.

184: So it has been studied already? Please quote the sources.

Response: We have modified this sentence into “the agarose-degrading activity and the agarase of V. natriegens are scarcely littile studied”. Please see line 190.

214: Bad sentence, please rephrase.

Response: The sentence has been revised into “In conclusion, V. natriegens WPAGA4 with agarose-degrading activity was isolated from the deep-sea sediments in the current work” (lines 231-233).

220: Why is cold adaptivity a positive trait in relation to industrial applications? Often industrial processes of viscous polymers favors high temperatures, where this enzyme would be ill suited. Please clarify.

Response: The cold-adapted enzyme possesses several advantages in the industrial application. The seasons are as follows:

Recently, the scientific interest in the cold-adapted enzymes is increasing. The cold-adapted enzymes could work at low and moderate temperatures (<40°C), thereby saving the costs of energy expenditures and equipment optimization in the heating process. Besides, the by-products in the reaction could be obviously reduced at the low temperatures. The cold-adapted rAga3420 in this work could serve as an energy-cost-saving bio-source for the agarose degradation and oligosaccharide production in the industry. Meanwhile, the low temperature could also prevent the caramelization, the potential by-product in high-temperature polysaccharide degradation. Therefore, the agarase rAga3420 with low temperature activity is not only potential driver of polysaccharide cycle but also an ideal bio-tool of the oligosaccharide production in the industrial application.

We have supplemented above content in lines 205-215.

244-267: All tools needs to be cited correctly with the appropriate papers and not only the URL.

Response: Thank you for the valuable comment, and the references have been provided in lines 267-282.

Reviewer 2 Report

Author characterised a deep-sea bacterium Vibrio-natriegens WPAGA4 as an agarose degrading bacteria and characterized the GH50 β-agarase. Although this work may provide the significance for the polysaccharides decomposition and production of NAOs, the overall manuscript presentation seems poor. Authors should deeply analyze the genome sequence for the presence of a single GH50 encoded Aga3420. The genome sequence should be analyzed for the detail CAZyme analysis before calculating the total Agarase activity by the strain. Authors mention “relatively” activity in Fig 2, and mention that the measurement was performed as described in our previous work. Author should at least mention, on what reference and how the relative activity was calculated. Line 318, author mention about the residual activity calculation and which is unclear to replicate the study by researcher, and never mention about the residual activity in the result and discussion section. Overall, the manuscript should be revised thoroughly for the consideration for review process. And The manuscript needs some major and minor corrections that are highlighted below and needs to be done.

Major corrections:

1.      In the manuscript, some additional work regarding the effect of salinity, effect of metal ions in the activity of enzymes can be done.

2.      Similar kind of work has been already done for example in Shewanella sp. WPAGa9 by Wenxin wang et al., 2021. How this study is different from that except the difference in strain? The study is based on the characterization of GH50 agarase from cold deep-sea environment. Is there any novelty of this work?

Minor corrections:

1.      Needs English corrections and more precise scientific writing.

2.      Change V. natriegens to Vibrio natriegens throughout the manuscript. Do italicize genus and species.

3.      Line 21, Abstract section “rAga3420 showed a cold-adapted property that 59.7% and 41.2% of the relative activity remained at 10°C and 0°C, respectively”. Please provide some of the references related to it in the Discussion section.

4.      The authors describe “The current work indicated that V. natriegens 70 WPAGA4 and the GH50 agarase possessed the application potential in the industrial production of NAOS and the ecological value for the polysaccharide decomposition and carbon cycle in the ocean, thereby widening our knowledge of the polysaccharide degradation of the deep-sea microorganisms” in the Introduction section, last sentence. It is inappropriate to write in this section. It should be omitted.

5.      Line 51, no need to have double space after fundamental of, only single space is enough.

6.      Line 100, italicize “V. natriegens” and make sure all the genus and species names of each microorganism are italicized throughout the manuscript.

7.      Line 106, “agarase production” please correct it to “agarose-degrading activity”

8.      Line 154 and 155, sentences are missing the references.

9.      Line 171-172, “As far as we know, the agarose concentration in the deep-sea environments has not been reported” does not correlate with this paragraph. Please correct it.

10.   Line 190 to 191, the sentence “Compared with the GH50 agarases from other environments, rAga3420 possessed outstanding cold-adapted property (Table S1)”. But Table S1 does not contain the isolation sites. Please correct this table.

11.   Line 232, “The single clone with agar collapse”, the authors here must have tried writing colony instead of clone, correct it.

12.   Line 242, 249, 250, 259, and 265, error because of the line spacing so it is better to correct.

13.   Line 270, please write like 25°C, 28°C, 31°C, 34°C instead of 25, 28, 31, 34. Similarly, in line no 316, writing temperature of 0°C, 10°C, and 20°C should be more appropriate.

14.   Line 312, please write the concentration of enzyme used rather than writing 10ul.

15.   Why didn’t the authors write about the protein concentration measurement? Please provide this information.

16.   Line 322, instead of writing Mg2+, Fe2+, Fe3+, Ga2+, Mn2+, Cu2+, Na+, K+, Zn2+, please use the superscript for writing the ions like Mg2+, Fe2+, Fe3+, Ga2+, Mn2+, Cu2+, Na+, K+, Zn2+.

17.   In the supplementary table, Table S1, In the first row of the table where optimal temperature is written, please remove one °C because that has been repeated.

18.   Please provide the DDH and ANI analysis values as a supplementary table. Which organisms were used for the analysis and was there any outgroup?

19.   In figure 1C, along with the Vibrio genus, insert closely related outgroups in the phylogenetic tree.

20.   To give more clarity and information, I request the authors to insert dbCAN result in the supplementary file and to determine/list out the agarose degrading genes that were obtained from the analysis. If possible, I request the authors to propose a pathway for degradation of agar based on genome analysis and the known reactions of the predicted enzymes related to agarose degradation in the current strain.

Author Response

Response: Thank you for the valuable comments. The manuscript has been revised according to your comments as follows: (1) The genome of the strain has been further analyzed to show the CAZyme genes’ family and amount (lines 114-121 and Fig. 1C); (2) The definition and the calculation method of “relative activity” have been provided in lines 349-351. The relative activity was defined as the percentage of activity determined with respect to the maximum agarase activity; (3) “residual activity” has been corrected into “relative activity” (line 343). We apologize for this mistake.

Other revisions also performed based on your major and minor comments. Please see the detail responses below.

In the manuscript, some additional work regarding the effect of salinity, effect of metal ions in the activity of enzymes can be done.

Response: Thank you for this comment. These experiments have been supplemented and the results have shown in the revised manuscript in lines 136, 137, 346, 347 and Figure 4G.

Similar kind of work has been already done for example in Shewanella sp. WPAGa9 by Wenxin wang et al., 2021. How this study is different from that except the difference in strain? The study is based on the characterization of GH50 agarase from cold deep-sea environment. Is there any novelty of this work?

Response: Thank you for the strict review. The major difference between the two works is that the products of rAga3420 (GH50 family) in this work is neoagarobiose (NA2), and that of Shewanella sp. WPAGA9 (GH16 family) are neoagarotetraose and neoagarohexaose. As we state in the introduction section, the GH50 agarase that can produce NA2 was rarely reported compared to GH16, especially the genus Vibrio. Our work studies a NA2-producing GH50 family from the deep-sea environment that could work at low temperature, revealing the industrial and ecological functions of this agarase and its host strain.

Minor corrections:

Needs English corrections and more precise scientific writing.

Response: Thank you for the strict review. We have corrected the writings of genus and species names, chemical formula, units, and the grammar in the manuscript. Please see lines 13, 15, 19-20, 23, 26, 58, 59, 61, 63, 73, 76, 78-79, 82-83, 94, 101, 103, 109, 112, 137, 162-163, 165-167, 170-173, 181-185, 188-191, 198, 202, 236, 299, 345-346, 348.

Change V. natriegens to Vibrio natriegens throughout the manuscript. Do italicize genus and species.

Response: The writings of genus and species have been corrected in lines 13, 15, 19-20, 23, 26, 58, 59, 61, 63, 73, 76, 78-79, 82-83, 94, 101, 103, 109, 112, 162-163, 165-167, 170-173, 181-185, 188-191, 198, 202, 236.

According to the standard binominal nomenclature in Species Plantarum [1] written by C Linnaeus, the founder of modern biological taxonomy, the genus name should be abbreviated when it appears again in the article. Thus, we humbly ask if the abbreviation could be kept in our manuscript to meet the requirement of the binominal nomenclature standard.

Reference

[1] Linnaeus C. Species plantarum[M]. Impensis GC Nauk, 1799.

Line 21, Abstract section “rAga3420 showed a cold-adapted property that 59.7% and 41.2% of the relative activity remained at 10°C and 0°C, respectively”. Please provide some of the references related to it in the Discussion section.

Response: Thank you for the valuable comment. The data has been discussed in lines 201-204.

The authors describe “The current work indicated that V. natriegens WPAGA4 and the GH50 agarase possessed the application potential in the industrial production of NAOS and the ecological value for the polysaccharide decomposition and carbon cycle in the ocean, thereby widening our knowledge of the polysaccharide degradation of the deep-sea microorganisms” in the Introduction section, last sentence. It is inappropriate to write in this section. It should be omitted.

Response: Thank you for the comment for improving our writing. The sentence has been removed. Please see lines 63-68.

Line 51, no need to have double space after fundamental of, only single space is enough.

Response: Thank you for the comment for improving our writing. The sentence has been corrected.

Line 100, italicize “V. natriegens” and make sure all the genus and species names of each microorganism are italicized throughout the manuscript.

Response: Thank you for the comment for improving our writing. The writing has been corrected. Please see lines 13, 15, 19-20, 23, 26, 58, 59, 61, 63, 73, 76, 78-79, 82-83, 94, 101, 103, 109, 112, 162-163, 165-167, 170-173, 181-185, 188-191, 198, 202, 236.

Line 106, “agarase production” please correct it to “agarose-degrading activity”

Response: Thank you for the comment for improving our writing. The writing has been corrected. Please see lines 100 and 101.

Line 154 and 155, sentences are missing the references.

Response: The references have been supplemented in lines 159-162.

Line 171-172, “As far as we know, the agarose concentration in the deep-sea environments has not been reported” does not correlate with this paragraph. Please correct it.

Response: Thank you for the comment. Until now, several agarose-degrading strains have been isolated from the deep-sea environment. However, the reason that the deep-sea strains obtained the agarose-degrading ability is still unexplored. The situ environment with agarose or agar could be possible reason, but the agarose concentration in deep sea is unmeasured. Therefore, what we want to express through this sentence is that, the current and former studies demonstrated the existence of the agarose-degrading strains in the deep sea, although the potential mechanisms are still unclear.

Line 190 to 191, the sentence “Compared with the GH50 agarases from other environments, rAga3420 possessed outstanding cold-adapted property (Table S1)”. But Table S1 does not contain the isolation sites. Please correct this table.

Response: The isolation sites have been supplemented in the Table S1.

Line 232, “The single clone with agar collapse”, the authors here must have tried writing colony instead of clone, correct it.

Response: The writing has been corrected in lines 250 and 251.

Line 242, 249, 250, 259, and 265, error because of the line spacing so it is better to correct.

Response: The format has been corrected in the revised manuscript.

Line 270, please write like 25°C, 28°C, 31°C, 34°C instead of 25, 28, 31, 34. Similarly, in line no 316, writing temperature of 0°C, 10°C, and 20°C should be more appropriate.

Response: The writing has been corrected in lines 290 and 291.

Line 312, please write the concentration of enzyme used rather than writing 10ul.

Response: The concentration of enzyme has been provided in line 335 (20.7 μg/mL).

Why didn’t the authors write about the protein concentration measurement? Please provide this information.

Response: The protein concentration was measured by using Bradford method. It has been supplemented in lines 331 and 332.

Line 322, instead of writing Mg2+, Fe2+, Fe3+, Ga2+, Mn2+, Cu2+, Na+, K+, Zn2+, please use the superscript for writing the ions like Mg2+, Fe2+, Fe3+, Ga2+, Mn2+, Cu2+, Na+, K+, Zn2+.

Response: The writing has been corrected in lines 345 and 346.

In the supplementary table, Table S1, In the first row of the table where optimal temperature is written, please remove one °C because that has been repeated.

Response: Table S1 has been revised according to your comment.

Please provide the DDH and ANI analysis values as a supplementary table. Which organisms were used for the analysis and was there any outgroup?

Response: The analysis results have been provided in Table S2.

In figure 1C, along with the Vibrio genus, insert closely related outgroups in the phylogenetic tree.

Response: This figure was produced using the online web server https://ggdc.dsmz.de/home.php by default. We cannot insert the outgroup in this website. However, we strongly agree with you that the outgroup is necessary for this analysis. Therefore, this figure was deleted from the result. Although the tree has been removed, the results of 16S rRNA alignment, DDH, and ANI can fully prove the taxonomy of this strain. Thank you very much for this strict review.

To give more clarity and information, I request the authors to insert dbCAN result in the supplementary file and to determine/list out the agarose degrading genes that were obtained from the analysis. If possible, I request the authors to propose a pathway for degradation of agar based on genome analysis and the known reactions of the predicted enzymes related to agarose degradation in the current strain.

Response: (1) The dbCAN result has been provided in the supplementary files (Fig. S2).

(2) The agarose degradation pathway, as your comment, has been reported by some former works, which indicates that its degradation needs a corporation of group enzymes. However, we failed to reveal other enzyme activities, including GH117 etc., thereby limiting us to speculate the degradation pathway. Nevertheless, we strongly agree with your point that the degradation pathway could be crucial in the current strain, and the related research will be performed in our future work.

Reviewer 3 Report

This is an interesting manuscript that isolates and investigates a novel agarase from a deep-sea Vibrio species.  The novelty lies in the thorough characterization of an agarase from a deep-sea environment, which has apparently not been done before.  As such, the work extends our knowledge of both agarases and deep-sea environments.  The authors note in particular the relatively high level of retention of activity at low temperatures, down to 0 degC, and the potential industrial applications.

The text is clearly written, and contains only minor grammatical errors that do not impinge on the reader's understanding of the meaning.  The figures are clearly presented, and the references seem thorough.  Only a small amount of content is present in the SI, which seems to not need any revision.

Some points:

1. Abstract and elsewhere "Kcat" is a rate constant, should be lower-case k

2. Intro line 3: "...D-galactose.... l-galactose...." be consistent, either use D- and L- or use d- and l-

3. Line 105 and elsewhere: CH4N2O.  Perhaps this is a standard formula for a compound, but I do not know what this is?  Even if I did, I could not be certain since there are many compounds with this formula.  Use a name if possible.

4. Fig. 2C: carbon source abbreviations are not defined in the caption or the text.

5. Section 2.4: should at least clarify that the over-expressed protein has a hexa-His tag.  This is clear in the experimental, but not mentioned in the main text.

6. Effect of EDTA and metal ions, etc.: Given the His tag, does this impact the results of the experiments with metals?  I am not familiar with this area, and perhaps it is well-established that His6 only binds nickel?  I also wonder whether nickel which will be bound would have an effect on the agarase?  Some clarification might be helpful here, to establish the relevance of the current results to non-tagged enzyme.

Overall, I think this is interesting work that goes slightly above being merely incremental, and with minor edits should be ready to publish in Marine Drugs.

Author Response

Response: Many thanks for your recognition to our work. We have revised the manuscript according to your valuable comment. Please see the detail responses below.

Some points:

Abstract and elsewhere "Kcat" is a rate constant, should be lower-case k

Response: The letter has been revised in lines 23 and 137. Thank you for this strict review.

Intro line 3: "...D-galactose.... l-galactose...." be consistent, either use D- and L- or use d- and l-

Response: The letter has been revised in line 34. Thank you for this strict review.

Line 105 and elsewhere: CH4N2O. Perhaps this is a standard formula for a compound, but I do not know what this is? Even if I did, I could not be certain since there are many compounds with this formula.  Use a name if possible.

Response: We are sorry for confusing you with the formula. CH4N2O is the formula of urea, and the name has been provided in the manuscript (lines 99, 106, and 299).

Fig. 2C: carbon source abbreviations are not defined in the caption or the text.

Response: We apologize for this mistake. The definitions of the carbon source abbreviations have been supplemented in lines 105-107.

Section 2.4: should at least clarify that the over-expressed protein has a hexa-His tag. This is clear in the experimental, but not mentioned in the main text.

Response: We have supplemented the information in line 126.

Effect of EDTA and metal ions, etc.: Given the His tag, does this impact the results of the experiments with metals? I am not familiar with this area, and perhaps it is well-established that His6 only binds nickel? I also wonder whether nickel which will be bound would have an effect on the agarase?  Some clarification might be helpful here, to establish the relevance of the current results to non-tagged enzyme.

Response: Usually, the His tag in the recombinant protein does not largely affect enzyme activities; however, we still have to emphasize the protein is recombinant instead of natural as the previous works [1-5]. Also, we mentioned that the characteristics is from the recombinant agarase Aga3420, namely rAga3420, rather than from the natural agarase.

  Nickel ion has the potential for impacting the enzyme activity. However, the nickel ion will be separate from the enzyme by the competitive binding of imidazole during the protein purification. Besides, the small molecular including nickel ion could be further removed by the dialysis process after the protein purification. Therefore, the nickel ion has little impact on the activity of rAga3420. This process has become a classify method for the activity determination of recombinant proteins (please also see the references [1-5]).

  At last, thank you again for all the valuable comments that greatly improve our work.

References

[1] Du C, Si Y, Pang N, et al. Prokaryotic expression, purification, physicochemical properties and antifungal activity analysis of phloem protein PP2-A1 from cucumber[J]. International Journal of Biological Macromolecules, 2022, 194: 395-401.

[2] Liao X, Wang W, Fan C, et al. Prokaryotic expression, purification and characterization of human cyclooxygenase-2[J]. International Journal of Molecular Medicine, 2017, 40(1): 75-82.

[3] Fu G, Cui Z, Huang T, et al. Expression, purification, and characterization of a novel methyl parathion hydrolase[J]. Protein expression and Purification, 2004, 36(2): 170-176.

[4] Cho C M H, Mulchandani A, Chen W. Bacterial cell surface display of organophosphorus hydrolase for selective screening of improved hydrolysis of organophosphate nerve agents[J]. Applied and Environmental Microbiology, 2002, 68(4): 2026-2030.

[5] Wierzbicka-Woś A, Bartasun P, Cieśliński H, et al. Cloning and characterization of a novel cold-active glycoside hydrolase family 1 enzyme with β-glucosidase, β-fucosidase and β-galactosidase activities[J]. BMC biotechnology, 2013, 13(1): 1-12.

Reviewer 4 Report

The current manuscript deals with the characterization of an original agarase. The whole manuscript is badly written, including many typos (species should be in italics, the chemical formulae are wrong,etc..). A major concern deals also with the unknown source of agarose (size, origin, mass, etc...), thus Michaelis Menten constants remains unclear. What's NA2? how the kinetics were performed if agarose was cut in many pieces? What's the enzymatic models? In addition the work should be better put in lights of previous studies, and compared to known agarases. In an overall manner the manuscript should be fully rewritten, especially the discussion section.

Author Response

Response: Thank you for the strict review for our manuscript. Please the detail responses below.

(1) All the species names have been corrected and written in italic, and the chemical formula of metal ions have also been revised. Please see lines 13, 15, 19-20, 23, 26, 58, 59, 61, 63, 73, 76, 78-79, 82-83, 94, 101, 103, 109, 112, 137, 162-163, 165-167, 170-173, 181-185, 188-191, 198, 202, 236, 299, 345-346, 348.

(2) The agarose in the work was purchased from WESTBIO company (Spanish). However, the molecular weight of agarose is very difficult to measure. Agarose is a polysaccharide and is a polymer with a molecular weight ranged from 80 000 and 140 000. Therefore, the molecular weight of agarose is large and variable. Nevertheless, the kinetic parameters are still measurable, because the agarases act on the glycosidic bonds instead of the whole agarose molecular. The determination method used in this work is a classify approach for the agarase kinetic parameters and have been used in many related works. Please see references [1-10].

(3) As we have stated in lines 51 and 52, NA2 is neoagarobiose, which is the product of the agarose degradation by the agarase in this work. According to the product result, the agarase in this work is exo-type, which means that its active model is cutting the agarose step by step and releasing NA2 from the end of the agarose molecular. Please see section 2.5 for the detail.

(4) This work aims to investigate the cold-adapted property of rAga3420, thereby revealing the potential industrial and ecological values of this agarase. In order to study this characteristic, we emphatically compared the properties of rAga3420 and other reported agarases in the discussion section and Table S1. Through the horizontal comparison, we found that rAga3420 possessed outstanding cold-adapted property and could serve as a potential tool for the oligosaccharide production. Please see lines 196-204 in the revised manuscript.

Finally, thanks a lot for all your constructive comments which largely improve our work and manuscript.

References

[1] Fu X T, Pan C H, Lin H, et al. Gene cloning, expression, and characterization of a $\beta $-agarase, AgaB34, from Agarivorans albus YKW-34[J]. Journal of microbiology and biotechnology, 2009, 19(3): 257-264.

[2] Di W, Qu W, Zeng R. Cloning, expression, and characterization of thermal‐stable and pH‐stable agarase from mangrove sediments[J]. Journal of basic microbiology, 2018, 58(4): 302-309.

[3] Chen Z W, Lin H J, Huang W C, et al. Molecular cloning, expression, and functional characterization of the β-agarase AgaB-4 from Paenibacillus agarexedens[J]. AMB Express, 2018, 8(1): 1-10.

[4] Temuujin U, Chi W J, Chang Y K, et al. Identification and biochemical characterization of Sco3487 from Streptomyces coelicolor A3 (2), an exo-and endo-type β-agarase-producing neoagarobiose[J]. Journal of bacteriology, 2012, 194(1): 142-149.

[5] Han Z, Zhang Y, Yang J. Biochemical characterization of a new β-agarase from Cellulophaga algicola[J]. International journal of molecular sciences, 2019, 20(9): 2143.

[6] Wang W, Wang J, Yan R, et al. Expression and Characterization of a Novel Cold-Adapted and Stable β-Agarase Gene agaW1540 from the Deep-Sea Bacterium Shewanella sp. WPAGA9[J]. Marine Drugs, 2021, 19(8): 431.

[7] Li R K, Ying X J, Chen Z L, et al. Expression and Characterization of a GH16 Family β-Agarase Derived from the Marine Bacterium Microbulbifer sp. BN3 and Its Efficient Hydrolysis of Agar Using Raw Agar-Producing Red Seaweeds Gracilaria sjoestedtii and Gelidium amansii as Substrates[J]. Catalysts, 2020, 10(8): 885.

[8] Li R K, Ying X J, Chen Z L, et al. Expression and Characterization of a GH16 Family β-Agarase Derived from the Marine Bacterium Microbulbifer sp. BN3 and Its Efficient Hydrolysis of Agar Using Raw Agar-Producing Red Seaweeds Gracilaria sjoestedtii and Gelidium amansii as Substrates[J]. Catalysts, 2020, 10(8): 885.

[9] Jeong Y J, Choi J W, Cho M S, et al. Isolation of Novel Exo-type β-Agarase from Gilvimarinus chinensis and High-level Secretory Production in Corynebacterium glutamicum[J]. Biotechnology and Bioprocess Engineering, 2019, 24(1): 250-257.

[10] Feng Z, Li M. Purification and characterization of agarase from Rhodococcus sp. Q5, a novel agarolytic bacterium isolated from printing and dyeing wastewater[J]. Aquaculture, 2013, 372: 74-79.

Round 2

Author Response

This study reports the isolation, genome sequencing and biochemical characterization of GH50 beta-agarase from a deep-sea marine bacterium. Overall, this study lacks depth of analysis and general significance and novelty of findings. The shallow analysis of the genome is a shame since a deeper analysis and proper genome comparison of close relatives could bring up several interesting characteristics. The choice to express and shallowly characterize only one enzyme, when there are at least three other putative GH50 in the genome is also detrimental to the study and should be revised. The existing analysis have some flaws (see comments below) and some experiments needs to be repeated.

Response: Thank you for the valuable comments. In our revised manuscript, the CAZymes in WPAGA4 were further analyzed, and a total of 173 carbohydrate-active enzymes genes were annotated in the genome of V. natriegens WPAGA4, including 16 auxiliary activity genes, 26 carbohydrate-binding modules, 15 carbohydrate ester genes, 75 glycoside hydrolase genes, 34 glyco-syltransferase genes, 7 polysaccharide lyase genes (Fig. 1C). Moreover, there are indeed other putative agarases in the genome; however, the recombinant proteins of these putative genes are found without any agarase activity. We contribute this result to a false annotation result; therefore, this data is not shown in the manuscript.

Response: If the authors indeed have expressed and attempted to characterize the remaining three GH50 genes in V. natriegens WPAGA4 or putative genes from other families in the genome, then this data have to be presented at least in supplementary. The annotation and sequence homologies to these are well within bounds for them to be GH50 agarases. If they indeed are not agarases and something else then that is a huge discovery and important, please provide proof.

Response: Thank you for your valuable advice. The primers and activity results of the remaining putative agarases have been supplemented in lines 125, 126, 325, and 326 and the supplementary materials (Figs. S3 and S4). As our previous response, they showed no agarase activity. Although this result is not ideal for our work, it is a usual situation for the protein activity verification studies, including the previous agarase researches [1-5]. The recombinant proteins often show different/no activities compared with their annotation results. For example, the work of Hess et al. published in Science [6] discovered lots of CAZyme genes from cow rumen by metagenomics and database annotation. Subsequently, the authors expressed 90 predicted genes in E.coli and cell-free system, but only 51 of 90 (57%) tested proteins showed the enzymatic activity. The authors contributed this inactivity of the remaining carbohydrate active candidates to a number of reasons, including false-positive prediction of carbohydrate-active enzyme domains, minimal expression and/or misfolding of candidate proteins, or suboptimal reaction conditions. This situation that several predicted agarase from the bacterial genome show no activity also usually happens in the agarase studies [1-5]. Thus it can be seen that finding the inactivity reason of predict genes is complicated, at least it required to optimize the annotation algorithm, database, and reaction condition. We agree with you that it is a very important research point, but providing the proof that why the predict genes show no activity is a challenging work for us and needs time to compete. We are sorry for unsatisfying your requirement, but now we can only give the possible reason that the putative genes show no activity in the discussion section (lines 236-241). Many thanks again for the comment and the inspiration for our future work.

References

  1. Chen ZW, Lin HJ, Huang WC, Hsuan SL, Lin JH, Wang JP. Molecular cloning, expression, and functional characterization of the β-agarase AgaB-4 from Paenibacillus agarexedens. AMB Express. 2018;8(1):49. Published 2018 Mar 28. doi:10.1186/s13568-018-0581-8
  2. Liu N, Mao X, Yang M, Mu B, Wei D. Gene cloning, expression and characterisation of a new β-agarase, AgWH50C, producing neoagarobiose from Agarivorans gilvus WH0801. World J Microbiol Biotechnol. 2014;30(6):1691-1698. doi:10.1007/s11274-013-1591-y
  3. Dong C, Lin B, Song Y, et al. Characterization and activity enhancement of a novel exo-type agarase Aga575 from Aquimarina agarilytica ZC1. Appl Microbiol Biotechnol. 2021;105(21-22):8287-8296. doi:10.1007/s00253-021-11553-y
  4. Liu N, Mao X, Du Z, Mu B, Wei D. Cloning and characterisation of a novel neoagarotetraose-forming-β-agarase, AgWH50A from Agarivorans gilvus WH0801. Carbohydr Res. 2014;388:147-151. doi:10.1016/j.carres.2014.02.019
  5. An K, Shi X, Cui F, et al. Characterization and overexpression of a glycosyl hydrolase family 16 beta-agarase YM01-1 from marine bacterium Catenovulum agarivorans YM01T. Protein Expr Purif. 2018;143:1-8. doi:10.1016/j.pep.2017.10.002
  6. Hess M, Sczyrba A, Egan R, et al. Metagenomic discovery of biomass-degrading genes and genomes from cow rumen. Science, 2011; 331(6016): 463-467. doi: 10.1126/science.1200387

13: This first sentence should be removed since it falsely advertises novelty without any concrete qualifications.

Response: We have revised this sentence into “The characterizations of GH50 agarases from Vibrio species have rarely been reported until now”. This statement is based on the records in the CAZy database (http://www.cazy.org/IMG/krona/GH50_krona.html). Please also see our introduction section (lines 58-60).

Response: I don’t agree with this statement. There are currently 4 Vibrio sp. GH50 agarases well characterized out of 25 total characterized members. This study also only provides one more so to suggest that this paper changes the landscape of the characterized Vibrio GH50 agarases is simply not true.

Response: We apologize that we did not clarify the statement clearly in the response. Compared with the GH16 agarases reported from Vibrio species, the Vibrio GH50 agarase are relatively few according to the CAZy database (http://www.cazy.org/IMG/krona/). In detail, 65 Vibiro species are recorded with GH16 genes in this database, but only 7 are recorded with GH50 genes. Please see the complete statement in our revised manuscript (lines 58-62), which is that “To date, GH16 agarases from Vibrio species (http://www.cazy.org/IMG/krona/GH16_krona.html) are well characterized, while the characteristics of other β-agarase families in genus Vibrio are rarely studies according to the records in the CAZy database (http://www.cazy.org/IMG/krona/GH50_krona.html) .”

  Besides, we have not and cannot state that this work changes the landscape of the characterized Vibrio GH50 agarases. As our statement, the Vibrio GH50 agarase is less studied than the Vibrio GH16, and this work obtained the Vibrio GH50 agarase; therefore, we study the strain and the related agarase for supplementing the researches on Vibrio GH50 agarase. Therefore, this work only aims to study the characteristics of this agarase to show the potential industrial and ecological functions of this strain and enzyme. Our revised manuscript clarified our aim in the abstract and discussion section, please see lines 26-29, and 242-250 for details.

37-40: This statement is hugely exaggerated and the reference that is quoted does not support it. This should be removed or rephrased.

Response: This sentence has been re-phrased into “Besides, the polysaccharides, including cellulose, chintin, and agarose, are the carbon sources in environments, and the degraders of these polysaccharides are also the participants in the carbon cycle [1]. Therefore, the degradation of agarose is of great value for the studies on industry production [1] and the carbon cycle [2,3]”, and the references that supported this statement has been supplemented. Please see lines 37-40.

Response: I don’t understand the selection or phrasing of the chosen polysaccharides. Yes these are found in the sea amongst many many other polysaccharides but what is the point with mentioning these three? And why is it relevant for industry? This whole paragraph does not make sense.

Response: Thank you for the question. As you mentioned, many polysaccharides are found in the ocean, we cannot list them all due to the space limitation. Therefore, we select the typical polysaccharides just like many papers [1-18], and agarose, as the polysaccharide we studied in this work, should be mentioned in the examples.

  This sentence is not independent in this paragraph of the manuscript. There are other sentences in this paragraph to explain the industrial potential. The complete paragraph is as follows: Agar is the main component in the cell wall of the red algae. As the skeleton structure of agar, agarose possesses the alternately repeated unit of β-D-galactose and 3,6-anhydro-L-galactose. The agarose can be degraded into oligosaccharides and monosaccharides with the potential for anti-inflammatory, anti-cancer, anti-caries, whitening, and moisturizing bio-activities in the industries of medicine, health products, cosmetics, and energy. Besides, the polysaccharides, including cellulose, chintin, and agarose, are the carbon sources in environments, and the degraders of these polysaccharides are also the participants in the carbon cycle. Therefore, the degradation of agarose is of great value for the studies on industry production and the carbon cycle (lines 33-42).

  We humbly think that this paragraph has its significance in the manuscript. The work aim is to study the agarases and the agarose-degrading strains; therefore, the significance and prospect of this aim should be mentioned in a paragraph.

References

  1. Yadav, Pranjali, et al. "Gold laced bio-macromolecules for theranostic application." International journal of biological macromolecules 110 (2018): 39-53.
  2. de Vos, Paul, et al. "Polymers in cell encapsulation from an enveloped cell perspective." Advanced drug delivery reviews 67 (2014): 15-34.
  3. Hicks, S. J., and R. J. Rowbury. "Virulence plasmid‐associated adhesion of Escherichia coli and its significance for chlorine resistance." Journal of Applied Bacteriology 61.3 (1986): 209-218.
  4. Prasad, Kamalesh, and Mukesh Sharma. "Green solvents for the dissolution and processing of biopolymers." Current Opinion in Green and Sustainable Chemistry 18 (2019): 72-78.
  5. Wang, Sen, et al. "Strength enhanced hydrogels constructed from agarose in alkali/urea aqueous solution and their application." Chemical Engineering Journal 331 (2018): 177-184.
  6. Seidi, Farzad, et al. "Synthesis of hybrid materials using graft copolymerization on non-cellulosic polysaccharides via homogenous ATRP." Progress in Polymer Science 76 (2018): 1-39.
  7. Kumar, Majeti NV Ravi. "A review of chitin and chitosan applications." Reactive and functional polymers 46.1 (2000): 1-27.
  8. Yoshimatsu, Miho, et al. "Biochemical characterization of cellulose-binding proteins (CBPA and CBPB) from the rumen cellulolytic bacterium Eubacterium cellulosolvens 5." Bioscience, biotechnology, and biochemistry 71.10 (2007): 2577-2580.
  9. Gupta, Kailash C., and MAJETI NV RAVI KUMAR. "An overview on chitin and chitosan applications with an emphasis on controlled drug release formulations." Journal of Macromolecular Science, Part C: Polymer Reviews 40.4 (2000): 273-308.
  10. Pradhan, S., A. K. Brooks, and V. K. Yadavalli. "Nature-derived materials for the fabrication of functional biodevices." Materials Today Bio 7 (2020): 100065.
  11. Alnoch, Robson Carlos, et al. "Recent trends in biomaterials for immobilization of lipases for application in non-conventional media." Catalysts 10.6 (2020): 697.
  12. Afewerki, Samson, et al. "Gelatin‐polysaccharide composite scaffolds for 3D cell culture and tissue engineering: towards natural therapeutics." Bioengineering & translational medicine 4.1 (2019): 96-115.
  13. Yadav, Pranjali, et al. "Gold laced bio-macromolecules for theranostic application." International journal of biological macromolecules 110 (2018): 39-53.
  14. Wynne, E. C., and J. M. Pemberton. "Cloning of a gene cluster from Cellvibrio mixtus which codes for cellulase, chitinase, amylase, and pectinase." Applied and environmental microbiology 52.6 (1986): 1362-1367.
  15. Latifi, Meriem, et al. "Chemical modification and processing of chitin for sustainable production of biobased electrolytes." Polymers 12.1 (2020): 207.
  16. Wang, Dingquan, et al. "Complete genome sequence of Microbulbifer sp. YPW1 from mangrove sediments in Yanpu harbor, China." Archives of Microbiology 203.10 (2021): 6143-6151.
  17. Liao, Jing, Bo Hou, and Huihua Huang. "Preparation, properties and drug controlled release of chitin-based hydrogels: An updated review." Carbohydrate Polymers (2022): 119177.
  18. Soorbaghi, Fatemeh Pashaei, et al. "Bioaerogels: Synthesis approaches, cellular uptake, and the biomedical applications." Biomedicine & Pharmacotherapy 111 (2019): 964-975.

118: The dbCAN2 server should be quoted correctly in the method section

Response: Good, but why is there a different approach described in line 275? “. The CAZyme genes were annotated by using CAZy database” Not sure what this means since the CAZydatabase doesn’t offer any annotation tools but merely information. Please elaborate

Response: We are terribly sorry for this mistake. All the CAZyme genes are annotated by dbCAN2 server. Please see lines 296 and 297 for our detail revision. Many thanks for this careful review.

Figure 3: The phylogenetic analysis method has not been described and is normally performed on amino acid sequences. Should also at least include all characterized sequences listed in the CAZy-database

Response: Thank you for the valuable comment. We have revised the phylogenetic analysis based on the amino acid sequence, and the method has been described. Please see lines 117-120 and 310-311 in the revised manuscript

Response: The task is simply to choose a subset of characterized members of the GH50 family including the characterized Vibrio GH50 aa sequences and perform a phylogenetic analysis of this. I have performed this hundreds of times and there’s certainly not computational bottleneck with 25 sequences. Furthermore phylogeny on nucleotide level between different families makes no sense and brings nothing in play. Please provide a proper amino acid sequences derived tree including branch lengths, so it is possible to assess the evolutionary and sequential distance to other characterized members of Vibrio GH50 members.

Response: We really thank you for this comment that improves the result and our understanding for phylogenetic analysis. We have revised the analysis, please see the revised Figure 3 in our manuscript.

134-135: The kinetic parameters needs standard deviations and the Michalis-Menten curves needs to be shown (at least in the supplementary files).

Response: The results has been revised in line 137-138 and the supplementary files (Fig. S1).

Response: There is only three substrate concentrations on the lineweaver Burk plot which does not constitute a proper Michaelis Menten behavior. Basically you can show anything you want with so few points. If the authors indeed have performed an accurate kinetic analysis and modelled a proper MichalisMenten behavior, then there should be several more data points available here. Please provide these. Furthermore there is no errorbars on the datapoints in the graph.

Response: Thank you for the careful and valuable review. We have supplemented more data and the error bars in this analysis. Please see lines 142-144, and Fig. S2.

184: So it has been studied already? Please quote the sources.

Response: We have modified this sentence into “the agarose-degrading activity and the agarase of V. natriegens are scarcely littile studied”. Please see line 190.

Response: Spelling error in “little” and that is not proper English to say scarcely little.

Response: Sorry for this careless. We have a copy error here, but the manuscript is correctly written in this location. Please see line 195. We apologize again for this mistake.

Finally, we appreciate all your comments which greatly improves our analysis and manuscript, and we also thank you for your valuable time to review our work. Humbly, we hope our revision could meet your requirement. Wish you all the best.

Reviewer 2 Report

Overall, the quality of the manuscript has improved.

Author Response

Thank you for all the comments and your valuable time for reviewing our work. Wish you all the best.

Reviewer 4 Report

The authors have answered correctly to all my comments.

Author Response

(The authors gave the same response as above.)

Round 3

Reviewer 1 Report

Thank you for the hard work preparing this manus. You have satisfied my inquires.

It is a shame that you could not detect specificity on the other GH50s in the genome. If I read it right you have truncated the proteins to contains only the catalytic domain which is a standard practise but be aware in the future about the exact coordinates for this truncation and whether it you are removing any important parts of the enzyme. dbCAN domain coordinates have at times considerable edge effects and will be too short.  Align with characterized domains or predict the 3D structure and design you truncation from this instead. Best of luck..